# Immersive Virtual Reality Therapy Is Supportive for Orthopedic Rehabilitation among the Elderly: A Randomized Controlled Trial

**DOI:** 10.3390/jcm12247681

**Published:** 2023-12-14

**Authors:** Justyna Mazurek, Błażej Cieślik, Adam Wrzeciono, Robert Gajda, Joanna Szczepańska-Gieracha

**Affiliations:** 1University Rehabilitation Centre, Wroclaw Medical University, 50-556 Wroclaw, Poland; 2Healthcare Innovation Technology Lab, IRCCS San Camillo Hospital, 30126 Venice, Italy; 3Faculty of Physiotherapy, Wroclaw University of Health and Sport Sciences, 51-612 Wroclaw, Poland; 4Department of Kinesiology and Health Prevention, Jan Dlugosz University in Częstochowa, 42-200 Częstochowa, Poland; 5Center for Sports Cardiology, Gajda-Med Medical Center in Pułtusk, 06-100 Pułtusk, Poland

**Keywords:** elderly, rehabilitation, physical activity, mood disorders, virtual reality exposure therapy

## Abstract

Objective: This research aimed to determine the efficacy of VR therapy in mitigating symptoms of depression, anxiety, and stress among older adults following arthroplasty surgery and to comprehend the influence of psychological improvement on changes in functional outcomes. Methods: Utilizing a parallel-group randomized controlled trial design, the study involved 68 osteoarthritis patients who had recently undergone either total hip or knee arthroplasty. Subjects were split into two groups. The experimental group underwent eight VR therapy sessions during their rehabilitation, while the control group was given standard care. Assessments encompassed both psychological and functional outcomes, with tools like the Hospital Anxiety and Depression Scale, Perceived Stress Scale, and the Barthel Index, among others. The experimental group showcased notable enhancements in both psychological and functional areas compared to the control group. Results: A significant (*p* value of < 0.001) relationship was found between psychological progress and functional recovery, indicating that psychological factors can serve as predictors for functional outcomes. Conclusions: The findings emphasize the promising role of VR therapy as a beneficial addition to the rehabilitation process for older adults’ post-hip and knee arthroplasty. The integration of psychological interventions in standard rehabilitation practices appears valuable, but further studies are needed to ascertain the long-term advantages of such an approach.

## 1. Introduction

Osteoarthritis (OA) is currently the most prevalent degenerative disease of the musculoskeletal system. It affects 53% of individuals aged 65+ globally, with a higher prevalence among women [1]. OA of the knee and hip is the third most common musculoskeletal disorder and is ranked as the 11th leading cause of disability worldwide [2]. OA gradually diminishes the patient’s ability to be self-reliant, move freely, and engage in social life, thus reducing the overall quality of life [3]. Individuals affected by OA are almost three times more likely to experience very severe pain, and twice as likely to suffer from high levels of psychological distress when compared to those without OA [4]. Many individuals with osteoarthritis experience depression, with a rate of occurrence at 20% [5]. Specifically, patients with both OA and depression have 38.8% higher direct medical costs than those who only have OA [6]. OA can also greatly affect a person’s health, resulting in chronic pain and functional limitations leading to decreased physical and mental well-being [7,8].

Similar biological pathways and neurotransmitters are responsible for both chronic pain and psychological impairment like depression and anxiety. Chronic pain is frequently the catalyst that prompts individuals with osteoarthritis to seek medical attention [9]. Chronic pain, depressed mood, stress, and anxiety often occur together, with depression being a frequent comorbidity of various chronic diseases, including OA [10,11]. When a patient has both conditions simultaneously, the effectiveness of standard treatment for each condition alone is decreased (e.g., individuals who have depression and anxiety report more severe pain, greater physical impairment, increased postoperative opioid use, and a lower rate of return to work than those without these mental health conditions) [12]. The considerable number of people affected by osteoarthritis in society results in a significant need for specialized treatment, which frequently surpasses the number of available appointments. As a result, the use of a waitlist is necessary. The public hospitals in Europe are facing long delays, which have been aggravated by the SARS-CoV-2 pandemic, resulting in extensive waiting time for specialist Orthopedic consultations [13]. This postponement in medical care can cause symptoms to worsen and pain to increase. Additionally, it can contribute to a deterioration in mental health (such as depression, stress, anxiety). This is also observed after undergoing a surgical procedure [14].

In the past few decades, joint replacement surgery has become the primary method of Orthopedic surgical treatment of advanced degenerative OA. This is especially true for hip and knee joints. Starting the rehabilitation process immediately after hip and/or knee replacement (total hip replacement—THR, total knee replacement—TKR) is crucial for recovery. Primary interventions for post-operative rehabilitation concentrate on exercises aimed at enhancing joint flexibility and strength, gait reeducation, and retraining functional abilities to optimize independence in tasks such as transferring (from bed to chair, using the toilet, exiting the shower/bath), personal hygiene (bathing and dressing), and broader daily activities [15]. Unfortunately, despite compelling evidence on the link between physical pain and mental well-being, psychological evaluation and intervention are not currently perceived as standard practice in Orthopedic rehabilitation in Europe. Even if all patients had access to the necessary resources to receive suitable treatment, there would still be unaddressed demand due to a shortage of professionally trained mental health clinicians.

Virtual reality (VR) is a technological concept that allows its users to experience the full immersion in a simulated world, providing them with a sense of actual presence through the use of multimodal stimuli [16,17]. The application of VR-based rehabilitation in Orthopedic surgery, especially in THR/TKR, has noticeably and significantly risen [18,19]. Several VR-based protocols have been proposed for rehabilitation following TKR and THR [16,18,19,20]. However, none of the articles examined the effectiveness of VR-based interventions for symptoms of stress, anxiety, and depression in patients with OA after THR or after TKR. As a result of this research gap, this study was conducted to investigate the impact of VR therapy on the severity of depressive symptoms, anxiety, and stress levels in OA patients who had undergone rehabilitation following THR or TKR. The research hypothesis was as follows: VR therapy has a beneficial effect on the mental and functional state of people undergoing rehabilitation after lower limb arthroplasty. The primary purpose of this study was to determine how effective VR therapy is in alleviating symptoms of depression and anxiety, as well as in reducing perceived stress level in older adults who have undergone arthroplasty surgery. Furthermore, the objective of the study was to evaluate the impact of psychological improvements on changes in functional outcomes.

## 2. Materials and Methods

### 2.1. Study Design and Setting

The study took place at St. Hedwig of Silesia Hospital in Trzebnica (Poland) and was designed as a parallel-group randomized controlled trial. Outcomes were evaluated at two intervals: before and after the intervention, with an outcome assessor who was blinded to the group assignments. Participants were evenly divided into two groups using the block randomization method. The sequence for randomization was generated using computer software, and participants were enrolled via sealed, sequentially numbered envelopes. This allocation process remained confidential until the participants were registered and assigned to their respective groups. An independent researcher oversaw the randomization to ensure that the assessors remained blinded throughout. While both the participants and those delivering the intervention were aware of their group assignments during the trial, there were no deviations from the planned intervention due to its context.

The study’s design adhered to the recommendations for phase three (VR3) of clinical trials utilizing VR in healthcare, emphasizing the efficacy of the proposed treatment compared to the control group [21]. The protocol was reviewed and approved by the Bioethics Committee at the Wroclaw Medical University (Wroclaw, Poland) under the reference number 119/KB/2023. The study was registered in the ClinicalTrials.gov database (NCT06002139). Participants provided written informed consent to partake in the research.

### 2.2. Participants

As illustrated in Figure 1, following an initial eligibility assessment, 68 participants were randomly allocated to one of two treatment groups. The experimental group (VR therapy group) consisted of 34 individuals. The remaining 34 participants constituted the control group (CON group). The inclusion criteria specified participants who were 60 years or older and had recently undergone hip or knee joint arthroplasty surgeries. The exclusion criteria included cognitive impairments hindering the independent completion of research questionnaires, recognized consciousness disorders, a history of bipolar affective disorder or other severe mental conditions, use of psychoactive drugs, ongoing psychiatric or individual psychological treatments, contraindications to virtual reality such as epilepsy, vertigo, or notable vision impairments, a functional status that restricts independent movement (although Orthopedic aids like crutches or walkers were acceptable), and refusal to partake in the study at any study stage.

### 2.3. Interventions

Both groups underwent a four-week conventional rehabilitation regimen that encompassed two hours of kinesiotherapy (120 min, inclusive of gait training), thirty minutes of ergotherapy, and three individualized physical therapy procedures such as laser therapy, magnetic therapy, and electrotherapy, all tailored to address specific ailments and requirements of each participant.

In addition to their regular treatments, the VR therapy group underwent eight sessions (20 min each, twice weekly) of immersive virtual reality therapy (VR therapy) using the VRTierOne device by Stolgraf^®^, Stanowice, Poland. This system employed VR HTC VIVE goggles (2017) and two controllers. The VRTierOne goal was to divert patients’ attention to a serene virtual environment, allowing relaxation and fostering recognition of their psychological strengths. Sessions began with the patient at a garden door (Figure 2A). As the door opened, the patient entered an evolving garden that became more vibrant each session. Midway, participants colored a mandala using the controllers (Figure 2B). The therapeutic impact of VRTierOne combined Erickson’s psychotherapy principles, calming music that grew more uplifting over time, cognitive engagement through mandala coloring, and the mood-enhancing green garden aesthetic rooted in Japanese design (Figure 2C). Therapy included metaphorical communication and posthypnotic therapeutic suggestions [22]. The music was crafted by a therapist–composer duo, and the immersive garden experience aimed to uplift spirits. Further details on the VRTierOne’s principles can be found in our prior work [23].

### 2.4. Outcome Measures

Both primary and secondary outcomes were administered at the beginning and again after a four-week treatment period. The primary outcome measures included the Hospital Anxiety and Depression Scale (HADS) and the Perceived Stress Scale (PSS-10). HADS is a 14-item self-report questionnaire designed to screen for anxiety (using the HADS-A subscale, which comprises seven items) and depression (using the HADS-D subscale, also with seven items) in patients in non-psychiatric settings. Both subscales have a cut-off point of 8/21. The Cronbach’s α for the scale ranges from 0.78 to 0.93, and the test-retest correlation stands at r = 0.80, as found by Bjelland et al. in 2002 [24]. The PSS-10 is a ten-item scale assessing the stress an individual perceived over the past month. Its questions are general, making it applicable to various subpopulations [25]. The items evaluate the perceived unpredictability, uncontrollability, and overload in respondents’ lives. Scores range from 0 to 40, with higher scores indicating greater perceived stress.

The secondary outcome measures included the Generalized Self-Efficacy Scale (GSES), Barthel Index (BI), Rivermead Mobility Index (RMI), Tinetti’s Short Scale, Short Physical Performance Battery (SPPB), the Perception of Stress Questionnaire (PSQ), and the Visual Analogue Scale (VAS) for pain assessment. The GSES is a ten-item psychometric scale assessing optimistic self-beliefs about coping with difficult demands [26]. It measures the confidence one has in addressing a wide variety of stressful or challenging situations. Scores on the GSES range from 10 to 40, with higher scores denoting stronger self-efficacy. The BI, an ordinal scale, evaluates performance in daily living activities, with each activity having an assigned point value [27]. With ten descriptive variables for daily activities and mobility, a higher BI score indicates a greater likelihood of post-hospitalization independent living. The RMI, derived from the Rivermead Motor Assessment Gross Function subscale, quantifies mobility impairment [28]. It comprises 14 items spanning from simple tasks like turning in bed to more demanding ones like running. Each item scores as either ‘unable’ (0) or ‘able’ (1), with a perfect score of 14 suggesting full mobility. Tinetti’s Short Scale, a condensed version of the Performance-Oriented Mobility Assessment (POMA), gauges a patient’s gait and balance. It rates patients based on tasks like moving from sitting to standing and maintaining an upright position for a set time, scored on a 3-point Likert scale. The SPPB combines results from tests on gait speed, chair stands, and balance. It yields a composite score between 0 (worst) and 12 (best), reflecting the overall physical functionality in elderly subjects [29]. Lastly, the PSQ by Plopa and Makarowski is a 27-item scale, with scores from 1 to 5 per item [30]. It assesses stress in areas such as emotional tension, external stress, and internal stress. Overall stress perception scores range from 21 to 105, with a score above 60 indicating elevated stress perception.

### 2.5. Data Analysis

The required sample size for this study was determined using the G*Power 3.1.9.4 software (Heinrich Heine University Düsseldorf, Germany) [31]. Based on the results of the primary outcome (the depression subscale of HADS) from our previous research on the elderly population, we anticipated an effect size of 0.25, equivalent to a partial eta squared of 0.06 [23]. Setting the significance level (*α*) at 0.05 and the statistical power (1 − *β*) at 0.95, a total of 54 participants was deemed necessary to achieve statistical significance. Taking into account an anticipated 25% dropout rate, we enrolled a sum of 68 participants.

Data were analyzed using JASP version 0.16.3 (University of Amsterdam, The Netherlands). Categorical variables were presented as frequency counts and percentages, while continuous variables were summarized using mean and standard deviation (SD). The Shapiro–Wilk test confirmed the normal distribution of the data. At the outset, baseline demographic variables were cross-compared between groups via unpaired t-tests (for continuous data) and *χ*^2^ tests (for categorical data). Intervention effects between groups (pre- vs. post-intervention) were examined using an analysis of variance (ANOVA), complemented by paired and unpaired *t* tests. The influence of psychological enhancement on functional progress was investigated through Spearman correlation and stepwise linear regression. A significance threshold was set at *α* < 0.05.

## 3. Results

### 3.1. Participant Characteristics

Out of 87 potential participants, 68 met the inclusion criteria and were randomized for the study. Three participants dropped out: two from the VR therapy group and one from the control group, all due to health complications requiring re-hospitalization (Figure 1). Table 1 shows that there were no statistically significant differences between the VR therapy and control groups at baseline.

### 3.2. Effectiveness of the Interventions

Table 2 presents the time x group interaction based on ANOVA, whereas Table 3 illustrates mean values, SD, and the between-group mean difference. The VR therapy group demonstrated significant improvements in their secondary outcomes compared to primary outcomes and compared to the control group. Specifically, for the HADS, the VRT group’s scores decreased by 57.8%, while the control group’s scores increased by 7.7%. This resulted in a between-group difference of −8.66 (*p* < 0.001). Further analysis of variance revealed a significant group × time interaction for HADS, indicated by an *F* value of 25.48, effect size *η*p^2^ of 0.29, and a *p* value of < 0.001. In the HADS-A assessment, the VRT group’s scores decreased by 62.5%, while those in the control group declined by 2.9%.

The pronounced difference between the groups was −4.37 (*p* < 0.001). ANOVA for HADS-A supported this with an effect size *η*p^2^ of 0.25 (*p* < 0.001). For the HADS-D, the VR therapy group’s scores decreased by 51.8%, whereas the control group increased 20.1%. This resulted in a significant between-group difference of −4.27 (*p* < 0.001). The group × time interaction for HADS-D was pronounced, with an *η*p^2^ of 0.19 (*p* < 0.001). Lastly, regarding the PSS-10, the VR group’s scores decreased by 11.2%, while those of the control group rose by 2.0%. The between-group difference for PSS-10 was −3.3 (*p* < 0.001), with a group × time interaction marked by an *F* value of 17.77 and effect size *η*p^2^ of 0.22 (*p* < 0.001).

A similar pattern emerged for secondary outcomes after treatment. The VR therapy group displayed marked benefits over the control group in both psychological and functional dimensions. In terms of psychological metrics, the VR group exhibited superior improvements: VAS by 2.6 points (*η*p^2^ = 0.32), GSES by 10.4 points (*η*p^2^ = 0.43), PSQ by 23.9 points (*η*p^2^ = 0.40), ES by 6.6 points (*η*p^2^ = 0.28), IS by 7.7 points (*η*p^2^ = 0.33), and ET by 9.7 points (*η*p^2^ = 0.39). In the functional domain, the VR group surpassed the control: Tinetti scores by 3.4 points (*η*p^2^ = 0.42), BI by 21.2 points (*η*p^2^ = 0.43), RMA-GF by 2.0 points (*η*p^2^ = 0.27), and SPPB by 3.8 points (*η*p^2^ = 0.39). All aforementioned differences were significant with *p* < 0.001.

### 3.3. Correlations and Predictors

Figure 3 depicts the linear correlation heatmap between the change in functional and change in psychological outcomes. All examined parameters showed a significant positive correlation, indicating that higher difference values in one outcome group corresponded to higher values in the other. Notably, significant correlations were observed between psychological parameter differences and SPPB differences (ranging from 0.47 to 0.61), BI (ranging from 0.31 to 0.56), and Tinetti (ranging from 0.27 to 0.61).

Table 4 outlines the psychological predictors of functional improvement based on stepwise regression results. For the change in BI, the change in HADS-A (*B* = 1.53) and the change in PSS-10 (*B* = 1.53) explain 37% of its variance. This model is significant with *F* = 18.39, *p* < 0.001. In the case of the change in Tinetti, the model explains 30% of its variance, primarily driven by changes in GSES (*B* = 0.12) and HADS-A (*B* = 0.17), with *F* = 13.25, *p* < 0.001. For the RMA change, the primary predictors are changes in GSES (*B* = 0.09) and PSS-10 (*B* = 0.15), accounting for 26% of its variance and yielding *F* = 10.67, *p* < 0.001. Finally, for the SPPB change, the key influencers are changes in GSES (*B* = 0.11) and PSS-10 (*B* = 0.23). This model explains 40% of the variance, with *F* = 13.46, *p* < 0.001.

## 4. Discussion

In recent years, numerous studies have substantiated the efficacy of VR in the context of broadly defined rehabilitation [19]. These studies focused mainly on functional results, i.e., mobility, strength, balance, or range of motion. However, there are no reports on the use of VR to improve the psychological aspects of Orthopedic patients’ health, which are important factors affecting the effectiveness of rehabilitation [32,33]. Building upon previous research that has established the efficacy of VR interventions in the treatment of psychiatric disorders, this study aimed to assess the effectiveness of VR therapy in alleviating symptoms of depression and anxiety, as well as in reducing perceived stress levels among older adults recovering from arthroplasty surgery.

Our results showed significant improvements in all examined psychological outcomes in the experimental group, while in the control group, the psychological status did not change and the HADS total score even increased significantly. These findings are consistent with our prior research across diverse rehabilitation domains. Employing the identical procedure within the experimental group, we identified a noteworthy enhancement in the patients’ psychological state [17,34]. Furthermore, within the study involving cardiac patients, we also identified a significant increase in the HADS total score, HADS-Anxiety score, and stress level score within the control group following rehabilitation [17]. These findings hold significant implications, highlighting the need to integrate psychological interventions alongside conventional rehabilitation programs.

Among the psychological outcomes assessed, we observed that perceived pain experienced a significant reduction in both groups; notably, the experimental group exhibited a significantly lower level of perceived pain. This study marks the first instance where a noteworthy reduction in pain was achieved within the group utilizing VR therapy. Among stroke patients, no significant reduction in pain was observed [34]. The observed correlation could be attributed to the interplay between pain perception and depression and anxiety. Higher levels of depression and anxiety disorders may lead to heightened pain sensitivity, thereby contributing to the relationship [35]. Pain exerts a considerable influence on the efficacy of rehabilitation programs. Notably, pain can serve as a potent motivational factor, driving patients to actively participate in their rehabilitation. When patients experience relief from or improvements in pain levels, they tend to exhibit greater adherence to prescribed exercises and therapies. Moreover, patients who effectively manage their pain often progress more expeditiously through their rehabilitation programs and are more inclined to express higher satisfaction levels with the quality of care they receive [36,37].

Another significant consideration pertains to the correlation between mental well-being and functional outcomes. Psychological disorders, encompassing conditions such as depression, anxiety, and stress, possess the capacity to exert influence across diverse dimensions of the recovery process. When characterized by heightened severity, these disorders can encumber the efficacy of rehabilitation, extend the recovery time, and reduce the quality of life experienced by patients [32,33,38,39]. Furthermore, numerous studies emphasize the imperative to identify and address mental health issues as an integral component of the rehabilitation process [40,41,42]. Therefore, the secondary objective of this study was to evaluate how psychological improvements influence changes in functional outcomes.

Our findings revealed significant enhancements in the functional domain within both study groups. Nevertheless, the results of a between-group comparison underline that the VR group exhibited a notably superior performance compared to the control group. Furthermore, the results indicated significant correlations, suggesting that greater differences in psychological outcomes corresponded to higher functional outcomes. In addition, psychological factors have been found to be significant predictors of functional improvements.

Depressive and anxiety disorders have been identified in the literature as potential factors that can impede the efficacy of rehabilitation programs [12]. Nevertheless, it is imperative to highlight the significance of self-efficacy, which was found to be an important determinant of rehabilitation program success [43,44,45]. Patients who believe in their ability to manage their condition, adhere to treatment plans, and achieve rehabilitation goals are more likely to experience positive outcomes (i.e., improved function, pain relief, and a higher quality of life). Additionally, patients with high self-efficacy may tend to experience less anxiety and depression as they believe they can overcome the challenges posed by their condition. It is noteworthy that our results revealed self-efficacy as a significant predictor of functional domain improvement in patients. This underscores the importance of integrating this often overlooked factor as an important parameter in the patient recovery process assessment [43].

Our study possesses certain limitations that warrant acknowledgment. Notably, the lack of a long-term outcome assessment in the context of recovery from arthroplasty surgery constitutes our primary limitation. This limitation may impact the comprehensiveness of our findings, particularly in assessing the enduring effects of the VR therapy over an extended period. Future research projects may benefit from incorporating follow-up outcome assessments to provide a more thorough understanding of the intervention’s efficacy and durability.

## 5. Conclusions

VR therapy emerges as a compelling therapeutic intervention for elderly patients recovering from arthroplasty surgery, that has the potential to be considered as part of standard treatment. The integration of VR therapy into conventional rehabilitation not only enhances patients’ psychological well-being but also fosters improved functional outcomes. The reduction of stress, anxiety, and pain, coupled with the enhancement of self-efficacy, serves as positive prognostic indicators for patients, both during their rehabilitation center stay and upon discharge. These factors are associated with an improved quality of life, increased self-reliance, and sustained physical improvement. Before treatment, patients often find themselves trapped in a vicious cycle where pain and limited mobility worsen their mental health, resulting in decreased physical activity. This, in turn, further deteriorates their functional status, quality of life, and long-term prognosis, potentially leading to permanent disability. The application of our proposed treatment holds the promise of breaking this cycle.

## Figures and Tables

**Figure 1 jcm-12-07681-f001:**
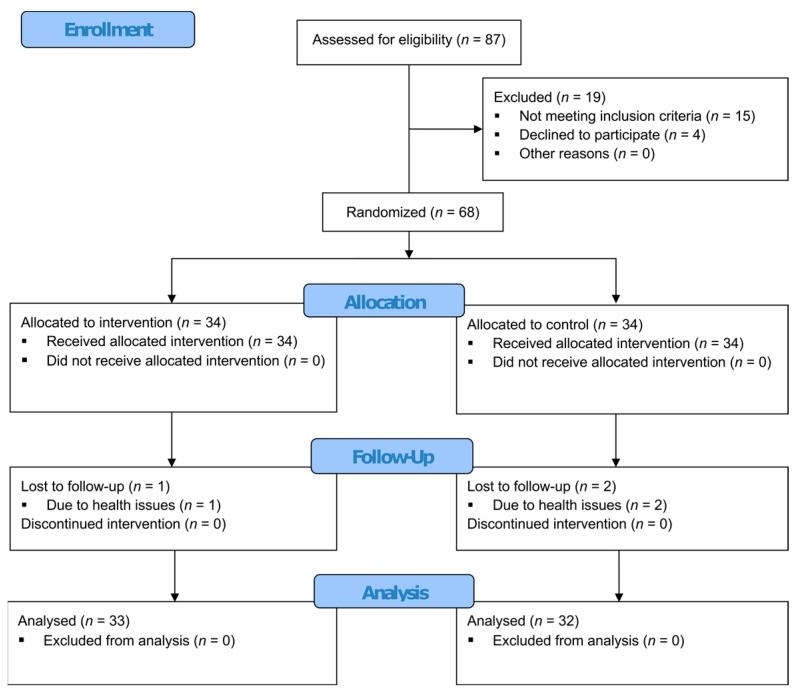
CONSORT study flow diagram.

**Figure 2 jcm-12-07681-f002:**
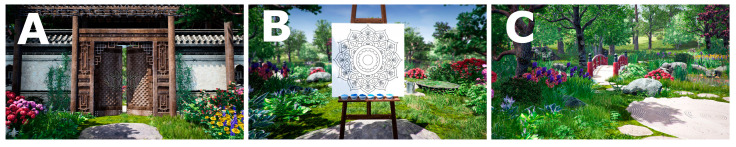
VRTierOne screenshots: (**A**) gate leading to the garden; (**B**) mandala coloring task; (**C**) decorative elements of the garden.

**Figure 3 jcm-12-07681-f003:**
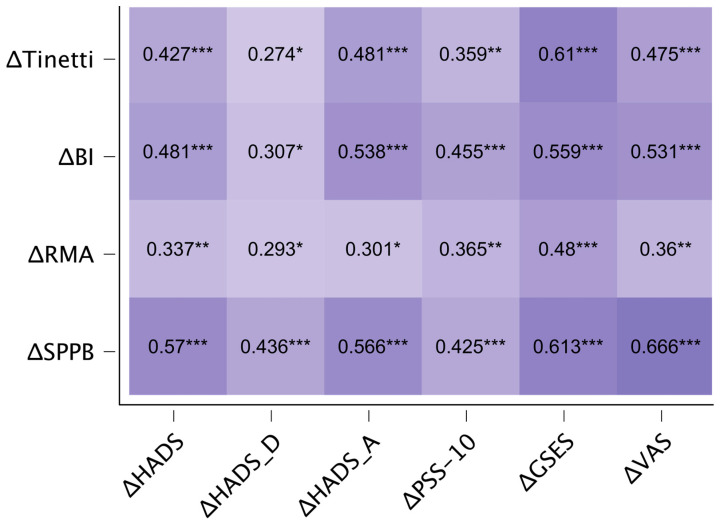
Correlation heatmap. * *p* < 0.05; ** *p* < 0.01; *** *p* < 0.001.

**Table 1 jcm-12-07681-t001:** Participants baseline characteristics.

Variable		Overall	VR Therapy	Control	*p* Value
*N*		68	34	34	-
*n* (%) of women		42 (61.76)	21 (61.76)	21 (61.76)	1.00 ^b^
Age, years		69.59 (6.16)	69.71 (6.82)	69.47 (5.52)	0.88 ^a^
Body mass, kg		80.25 (16.08)	79.26 (16.60)	81.24 (15.72)	0.62 ^a^
Body height, cm		167.69 (10.02)	166.76 (8.07)	168.62 (11.79)	0.45 ^a^
Body mass index, kg/m^2^		28.54 (5.12)	28.46 (5.25)	28.62 (5.05)	0.89 ^a^
	Normal (BMI 18.5–24.9), *n* (%)	13 (19.12)	6 (17.65)	7 (20.59)	0.76 ^b^
	Overweight (BMI 25–29.9), *n* (%)	33 (48.53)	18 (52.94)	15 (44.12)	0.47 ^b^
	Obese (BMI > 30), *n* (%)	22 (32.35)	10 (29.41)	12 (35.29)	0.60 ^b^
Arthroplasty area, *n* (%)					
	Hip	45 (66.18)	22 (64.70)	23 (67.65)	0.80 ^b^
	Knee	23 (33.83)	12 (35.30)	11 (32.35)	
Marital status, *n* (%)					
	Married	46 (67.65)	21 (61.76)	25 (73.53)	0.30 ^b^
	Single	1 (1.47)	1 (2.94)	0 (0.00)	-
	Widowed	21 (30.88)	12 (35.29)	9 (26.47)	0.43 ^b^
Education, *n* (%)					
	Primary/vocational	27 (39.71)	13 (38.24)	14 (41.18)	0.80 ^b^
	Secondary	29 (42.65)	16 (47.06)	13 (38.24)	0.46 ^b^
	Higher	12 (17.65)	5 (14.71)	7 (20.59)	0.52 ^b^

VR: virtual reality; BMI: body mass index; ^a^ *t*-test; ^b^ Chi-square test.

**Table 2 jcm-12-07681-t002:** ANOVA results (time × group).

Outcome	Mean Square	*F*	*η*p^2^	*p* Value
Psychological outcomes				
HADS	605.87	25.48	0.29	<0.001
HADS-A	156.29	21.23	0.25	<0.001
HADS-D	146.72	14.79	0.19	<0.001
VAS	53.95	30.88	0.32	<0.001
PSS-10	94.67	17.77	0.22	<0.001
GSES	837.89	47.85	0.43	<0.001
PSQ	4563.31	41.26	0.40	<0.001
ES	345.40	23.88	0.28	<0.001
IS	452.43	30.83	0.33	<0.001
ET	767.11	41.03	0.39	<0.001
Functional outcomes				
Tinetti	92.01	44.79	0.42	<0.001
BI	3520.84	47.48	0.43	<0.001
RMA-GF	35.19	23.74	0.27	<0.001
SPPB	111.54	41.03	0.39	<0.001

HADS: Hospital Anxiety and Depression Scale; HADS-A: anxiety subscale of the HADS; HADS-D: depression subscale of the HADS; VAS: visual analogue scale; PSS-10: Perceived Stress Scale; GSES: General Self-Efficacy Scale; PSQ: Perception of Stress Questionnaire; ES: External Stress; IS: Internal Stress; ET: Emotional Tension; BI: Barthel Index; RMA-GF: Rivermead Motor Assessment Gross-Function; SPPB: Short Physical Performance Battery.

**Table 3 jcm-12-07681-t003:** Mean values (SD) of primary and secondary outcomes.

Outcome	VR Therapy (*n* = 34)			Control (*n* = 34)			Between-Group Comparison	
	Baseline	Post-Treatment	*p* Value	Baseline	Post-Treatment	*p* Value	Mean Difference	*p* Value
Psychological outcomes								
HADS	13.15 (5.94)	5.55 (4.64)	<0.001	13.88 (7.68)	14.94 (7.00)	0.02	−8.66	<0.001
HADS-A	7.35 (3.33)	2.76 (2.50)	<0.001	7.56 (3.92)	7.34 (4.04)	0.72	−4.37	<0.001
HADS-D	5.79 (2.59)	2.79 (3.13)	0.004	6.32 (4.35)	7.59 (4.14)	0.06	−4.27	<0.001
VAS	5.27 (1.97)	0.88 (1.02)	<0.001	4.35 (2.07)	2.59 (1.94)	<0.001	−2.63	<0.001
PSS-10	24.94 (3.59)	22.15 (1.77)	<0.001	25.18 (3.77)	25.69 (4.27)	0.14	−3.30	<0.001
GSES	28.59 (6.62)	38.70 (1.29)	<0.001	30.74 (5.71)	30.41 (6.36)	0.88	10.44	<0.001
PSQ	59.32 (18.47)	35.73 (7.71)	<0.001	54.59 (18.64)	54.91 (15.08)	0.94	−23.91	<0.001
ES	18.06 (6.33)	12.73 (3.38)	<0.001	16.47 (6.03)	17.69 (4.97)	0.09	−6.55	<0.001
IS	19.56 (6.65)	11.30 (2.84)	<0.001	18.94 (7.09)	18.38 (6.01)	0.42	−7.70	<0.001
ET	21.71 (7.20)	11.70 (3.85)	<0.001	19.18 (6.52)	18.84 (5.41)	0.69	−9.67	<0.001
Functional outcomes								
Tinetti	3.29 (2.42)	9.64 (0.96)	<0.001	3.59 (2.50)	6.53 (2.27)	<0.001	−3.41	<0.001
BI	54.56 (16.49)	93.94 (6.47)	<0.001	56.18 (17.41)	74.38 (16.84)	<0.001	−21.18	<0.001
RMA-GF	4.56 (2.35)	10.00 (1.00)	<0.001	4.41 (2.55)	7.81 (2.47)	<0.001	−2.04	<0.001
SPPB	2.62 (2.34)	9.12 (2.41)	<0.001	2.88 (2.43)	5.59 (2.98)	<0.001	−3.79	<0.001

HADS: Hospital Anxiety and Depression Scale; HADS-A: anxiety subscale of the HADS; HADS-D: depression subscale of the HADS; VAS: visual analogue scale; PSS-10: Perceived Stress Scale; GSES: General Self-Efficacy Scale; PSQ: Perception of Stress Questionnaire; ES: External Stress; IS: Internal Stress; ET: Emotional Tension; BI: Barthel Index; RMA-GF: Rivermead Motor Assessment Gross-Function; SPPB: Short Physical Performance Battery; SD: standard deviation.

**Table 4 jcm-12-07681-t004:** Psychological predictors for functional improvement (stepwise regression results).

Variable		*B*	Beta	*t*	*p* Value	*F*	*R^2^*
ΔBI					<0.001	18.39	0.37
	ΔHADS-A	1.53	0.42	4.06			
	ΔPSS-10	1.53	0.35	3.38			
ΔTinetti					<0.001	13.25	0.30
	ΔGSES	0.12	0.36	2.96			
	ΔHADS-A	0.17	0.28	2.33			
ΔRMA-GF					<0.001	10.67	0.26
	ΔGSES	0.09	0.34	2.88			
	ΔPSS-10	0.15	0.28	2.42			
ΔSPPB					<0.001	13.46	0.40
	ΔGSES	0.11	0.29	2.47			
	ΔPSS-10	0.23	0.29	2.69			
	ΔHADS-A	0.18	0.26	2.32			

HADS-A: anxiety subscale of the HADS; PSS-10: Perceived Stress Scale; GSES: General Self-Efficacy Scale; RMA-GF: Rivermead Motor Assessment Gross-Function; SPPB: Short Physical Performance Battery.

## Data Availability

Data are available upon reasonable request to the corresponding author.

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
