# Peer review of "Immersive Virtual Reality Therapy Is Supportive for Orthopedic Rehabilitation among the Elderly: A Randomized Controlled Trial"

_jcm, 2023, doi:10.3390/jcm12247681_

Round 1
Reviewer 1 Report
Comments and Suggestions for Authors
Totally speaking, via determining the efficacy of VR therapy in mitigating symptoms of depression, anxiety, and stress among older adults following arthroplasty surgery and to comprehend the influence of psychological improvement on changes in functional outcomes, the present study aims to discuss whether immersive virtual reality therapy could be useful for orthopaedic rehabilitation of the elderly by a randomized controlled trial. No doubt, the relevant results from the present work have notable practical values, and can be also employed in clinical practice to help older people recover from this issue as fast as possible. The research design is proper, and research process is valid, statistical analyses are accurate, and the research results are reliable and persuasive, and therefore, this manuscript can be taken into account to receive for the present journal, but some detailed issues should be resolved by authors in advance, which are listed as follows:
1. The title can be revised as “Immersive Virtual Reality Therapy is Supportive for Orthopaedic Rehabilitation among the elderly: A Randomized-Controlled Trial”.
2. The abstract section should be clearly displayed with structural form, with objective, methods, results, conclusions.
3. Table 1 and table 2 has been cross-paged, and the significance icon in table 2, namely “*”, should be deleted, because the specific P value has been listed here. In addition, for table 4, the symbol R should be presented with italics.
4. The section of conclusion should be further refined and compressed.
5.The expression of references could be also carefully checked again!
That’s all, and thank you!
Comments on the Quality of English LanguageModerate editing of English language could be required for authors.
Author Response
Response to Reviewer #1 comments
Dear Reviewer,
Thank you for your insightful review of our manuscript. We greatly appreciate each of the comments you have made, as they have enriched the article, making it more interesting and accessible to the reader. As a result of the reviewer's feedback, we have corrected the abstract, supplemented the introduction with the hypothesis, reorganized the results section, and changed the conclusions section. Moreover, the title has been changed and now reads: "Immersive Virtual Reality Therapy Is Supportive for Orthopaedic Rehabilitation Among the Elderly: A Randomized-Controlled Trial”. All in-text changes are marked in red as requested by the journal guidelines for revision. We firmly believe that the current version of the article meets your expectations and is suitable for publication in Journal of Clinical Medicine.
Yours sincerely,
Authors
--
Totally speaking, via determining the efficacy of VR therapy in mitigating symptoms of depression, anxiety, and stress among older adults following arthroplasty surgery and to comprehend the influence of psychological improvement on changes in functional outcomes, the present study aims to discuss whether immersive virtual reality therapy could be useful for orthopaedic rehabilitation of the elderly by a randomized controlled trial. No doubt, the relevant results from the present work have notable practical values, and can be also employed in clinical practice to help older people recover from this issue as fast as possible. The research design is proper, and research process is valid, statistical analyses are accurate, and the research results are reliable and persuasive, and therefore, this manuscript can be taken into account to receive for the present journal, but some detailed issues should be resolved by authors in advance, which are listed as follows.
Authors response: We are grateful for the positive and constructive comments that originated in the review process. We have carefully reviewed the comments and have revised the manuscript accordingly.
The title can be revised as “Immersive Virtual Reality Therapy is Supportive for Orthopaedic Rehabilitation among the elderly: A Randomized-Controlled Trial”.
Authors response: Thank you, we have revised the title according to the suggestion.
The abstract section should be clearly displayed with structural form, with objective, methods, results, conclusions.
Authors response: We have revised the abstract.
Table 1 and table 2 has been cross-paged, and the significance icon in table 2, namely “*”, should be deleted, because the specific P value has been listed here. In addition, for table 4, the symbol R should be presented with italics.
Authors response: Thank you for your valuable comments that we have added to the article.
The section of conclusion should be further refined and compressed.
Authors response: We have refined and compressed our conclusions.
The expression of references could be also carefully checked again!
Authors response: We have revised the references.
That’s all, and thank you!
Authors response: Thank you!

Reviewer 2 Report
Comments and Suggestions for Authors
Dear authors,
Thank you for the effort you put into your research. The research is valuable and original in terms of subject and scope. After the corrections I will give below, I think your research is acceptable.
Abstract
Revise this section by dividing it into sections (background, method, results, conclusion) and make the results section more detailed (highlight the p values).
Introduction
This section was written very fluently, the only problem is that you talked about your topic and VR very briefly. I recommend getting down to the purpose by combining your topic and VR by combining the two paragraphs before your purpose sentence.
Please give your hypothesis after the purpose statement.
I think the method and results section is sufficient. I would just recommend that you make your tables more neat and understandable.
Discussion
It seems like you discussed few references in this section. Try to get support from references where you can give more detailed information about the subject (if you think the reference is sufficient, just explain the reason).
Best
Author Response
Dear Reviewer,
Thank you for your insightful review of our manuscript. We greatly appreciate each of the comments you have made, as they have enriched the article, making it more interesting and accessible to the reader. As a result of the reviewer's feedback, we have corrected the abstract, supplemented the introduction with the hypothesis, reorganized the results section, and changed the conclusions section. Moreover, the title has been changed and now reads: "Immersive Virtual Reality Therapy Is Supportive for Orthopaedic Rehabilitation Among the Elderly: A Randomized-Controlled Trial”. All in-text changes are marked in red as requested by the journal guidelines for revision. We firmly believe that the current version of the article meets your expectations and is suitable for publication in Journal of Clinical Medicine.
Yours sincerely,
Authors
--
Thank you for the effort you put into your research. The research is valuable and original in terms of subject and scope. After the corrections I will give below, I think your research is acceptable.
Authors response: Thank you very much for appreciation of our efforts and constructive feedback.
Abstract: Revise this section by dividing it into sections (background, method, results, conclusion) and make the results section more detailed (highlight the p values).
Authors response: Thank you for your suggestion. We have corrected abstract according to your comment.
Introduction: This section was written very fluently, the only problem is that you talked about your topic and VR very briefly. I recommend getting down to the purpose by combining your topic and VR by combining the two paragraphs before your purpose sentence. Please give your hypothesis after the purpose statement.
Authors response: We agree with the Reviewer – we have corrected the Introduction part.
I think the method and results section is sufficient. I would just recommend that you make your tables more neat and understandable.
Authors response: We have corrected the tables and reorganized the results section.
Discussion: It seems like you discussed few references in this section. Try to get support from references where you can give more detailed information about the subject (if you think the reference is sufficient, just explain the reason).
Authors response: Thank you for your comment. If you agree with this, we believe that the literature included in the discussion is sufficient for two reasons: we tried to select the latest research for the topic and meta-analysis, and also due to certain limitations in the amount of cited literature imposed by the journal's editors.
Best
Authors response: The same to you!

Reviewer 3 Report
Comments and Suggestions for Authors
Q1. In their experiment, the VR group compared to the control group experienced additional 8 times VR therapy, this may cause one alternative possibility: the improvement of VR group psychological status is because they experience VR scenario, no matter this specific scenario is about "Rehabilitation", I mean, you may achieve the same results only let them to play some VR game, if so why you put so much effort to design this rehabilitation scenario? How to exclude this possibility? I think you need to run the control experiment as I also experience 8 times VR experience, but this experience is that participants just need to wander in the environment, or do some routine things.
Q2. On line 224, the authors wrote "in their primary outcome", this may be misleading, as I understand, the primary and secondary outcomes(line 157) correspond to baseline and post treatment in Table 2, thus on line 224,it may be better to write " significant improvements in their second as compared to primary outcomes"?
Q3. I know usually we run simple comparison between each two conditions, only after we find significant interactions, thus I think the results in Table 2 and 3 should be rearranged, that is, the authors should present time and group main effect, also the interaction effect first, and then present the detailed comparison between the two times in each group, and also present the two group baseline comparison results, so to let the readers know that there was no baseline difference between the two groups.
Author Response
Dear Reviewer,
Thank you for your insightful review of our manuscript. We greatly appreciate each of the comments you have made, as they have enriched the article, making it more interesting and accessible to the reader. As a result of the reviewer's feedback, we have corrected the abstract, supplemented the introduction with the hypothesis, reorganized the results section, and changed the conclusions section. Moreover, the title has been changed and now reads: "Immersive Virtual Reality Therapy Is Supportive for Orthopaedic Rehabilitation Among the Elderly: A Randomized-Controlled Trial”. All in-text changes are marked in red as requested by the journal guidelines for revision. We firmly believe that the current version of the article meets your expectations and is suitable for publication in Journal of Clinical Medicine.
Yours sincerely,
Authors
--
Q1. In their experiment, the VR group compared to the control group experienced additional 8 times VR therapy, this may cause one alternative possibility: the improvement of VR group psychological status is because they experience VR scenario, no matter this specific scenario is about "Rehabilitation", I mean, you may achieve the same results only let them to play some VR game, if so why you put so much effort to design this rehabilitation scenario? How to exclude this possibility? I think you need to run the control experiment as I also experience 8 times VR experience, but this experience is that participants just need to wander in the environment, or do some routine things.
Authors response: Thank you for raising an interesting point. We appreciate your thoughtful consideration of potential alternative explanations. In our study, the design of the VR scenario was specifically tailored to integrate therapeutic elements based on the principles of Erickson’s psychotherapy.
This is the first stage of research where we assess the effectiveness of the VR intervention compared to standards of care. The same study design was used earlier, during studies among patients with various diseases, including neurological, cardiological, pulmonological and geriatric diseases. Orthopedic patients are another large group of patients where we focus on assessing the unique impact of a therapeutic VR scenario designed for rehabilitation purposes.
Your suggestion is worth serious consideration in the planning of future experiments. We appreciate your valuable input in enhancing the robustness of our research. In the next stages of the research, we plan to explore control experiments where participants will experience VR scenarios unrelated to rehabilitation, such as generic VR games or routine environments, as well as other psychological therapies. This approach could help us distinguish between the general effects of VR exposure and the targeted therapeutic benefits of our rehabilitation-focused VR scenario.
Q2. On line 224, the authors wrote "in their primary outcome", this may be misleading, as I understand, the primary and secondary outcomes (line 157) correspond to baseline and post treatment in Table 2, thus on line 224, it may be better to write " significant improvements in their second as compared to primary outcomes"?
Authors response: We have changed the sentence (now line 231-232).
Q3. I know usually we run simple comparison between each two conditions, only after we find significant interactions, thus I think the results in Table 2 and 3 should be rearranged, that is, the authors should present time and group main effect, also the interaction effect first, and then present the detailed comparison between the two times in each group, and also present the two group baseline comparison results, so to let the readers know that there was no baseline difference between the two groups.
Authors response: Thank you very much for that comment. We have reorganized the results section according to the reviewer’s suggestion.

Round 2
Reviewer 3 Report
Comments and Suggestions for Authors
no more comments.